# MultiCMET: A Novel Chinese Benchmark for Understanding Multimodal Metaphor

**Dongyu Zhang[1,2], Jingwei Yu[1], Jinsen Yuan[1], Liang Yang[3], Hongfei Lin[3*]**
[1] School of Software, [2]School of Foreign Languages,
[3]School of Computer Science and Technology, Dalian University of Technology, China

## Abstract

Metaphor is a pervasive aspect of human communication, and its presence in multimodal forms has become more prominent with the progress of mass media. However, there is a scarcity of research focusing on multimodal metaphor resources in languages other than English, despite the evident variations in how different languages express multimodal metaphors. Furthermore, the existing work in natural language processing does not address the exploration of categorizing the source and target domains in metaphors. This omission is significant considering the extensive research conducted in the fields of cognitive linguistics, which emphasizes that a profound understanding of metaphor relies on recognizing the differences and similarities between domain categories. We, therefore, introduce MultiCMET, a multimodal Chinese metaphor dataset, consisting of 13,820 text-image pairs of advertisements with manual annotations of the occurrence of metaphors, source and target domain categories, and sentiments metaphors convey. Furthermore, we have developed a domain lexicon that encompasses categorizations of domains and corresponding examples. We propose the Cascading Domain Knowledge Integration (CDKI) benchmark, which utilizes domain-specific features from the domain lexicon to enhance the understanding of metaphors. Experimental results demonstrate the effectiveness of CDKI. The dataset and code are publicly available. [1]

## 1 Introduction

Metaphor plays an important role in human cognition and communication, and its ubiquity has been established by empirical studies showing the occurrence of metaphor about once in every three sentences in regular parlance (Steen et al., 2010; Shutova et al., 2010). With the development of

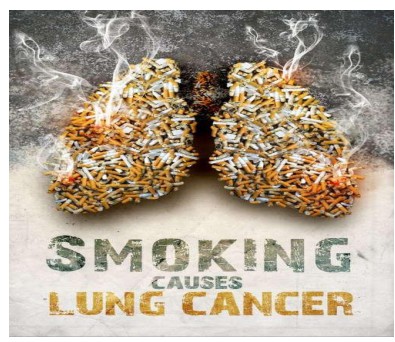

Figure 1: Example of Metaphor

modern media, when compared to monomodal metaphors, multimodal metaphors are significantly increasing due to their vivid, attractive, and persuasive effects. A multimodal metaphor is defined as a mapping that conceptualizes one target domain in terms of another source domain from different modes, such as text and image, text and sound, or image and sound (Forceville and Urios-Aparisi, 2009). Figure 1 presents a compelling metaphorical representation depicting lungs constructed from cigarettes, symbolizing the connection between two distinct entities, namely the lung and the cigarette. This metaphor invokes the perceptual notion that smoking is a leading cause of lung cancer. The source domain, represented by the "cigarette" image, intertwines with the target domain, represented both textually and visually through the depiction of the "lung". Understanding multimodal metaphor mainly focuses on identification of the underlying mapping of the two domains as well as the extraction of properties conveyed by metaphors(Kintsch, 2000; Su et al., 2017). It also requires the decoding of implicit messages and the identification of the semantic relationship between target and source domains (Coulson and Van Petten, 2002; Yang et al., 2013), so it involves cognitive efforts, which makes it significantly challenging for machines.

In recent years, there has been a surge of interest

---

*Corresponding author
[1]https://github.com/PedaloJSY/MultiCMET

in multimodal metaphor research within the field of Natural Language Processing (NLP) (Shutova et al., 2016; Xu et al., 2022; Ying et al., 2023). However, the study of multimodal metaphors in the Chinese language is extremely limited. It is important to note that metaphorical linguistic and visual expressions may vary across languages and cultures, despite the existence of universal conceptual metaphors (Kövecses, 2010). For example, "human is animal" is a universal conceptual metaphor in which the attributes or behaviors of animals are used to describe human attributes or behaviors. However, various languages may pick up different metaphorical linguistic and visual expressions underlying the same or similar semantics to realize the metaphorical conceptualization. For example, to use a metaphor to describe the state of being drunk, English speakers choose newt or skunk as a source domain, such as "as drunk as a skunk", while Chinese speakers would choose mud, as in "as drunk as mud". Due to the variations in multimodal metaphors across different languages, there is an urgent need to construct multimodal metaphor datasets in languages other than English. This will enable a more comprehensive automatic understanding of metaphor.

Furthermore, previous research on multimodal metaphor understanding has been limited in its depth of analysis, primarily focusing on metaphor detection tasks while neglecting tasks related to deep metaphor comprehension. These tasks include differentiating various categories of source domains and target domains, as they significantly contribute to the emergence of metaphors and facilitate a more profound understanding of their underlying meanings.Therefore, given the importance of domain classification knowledge, the establishment of a domain dictionary that includes domain categorization is crucial for the automatic comprehension of metaphors.

To bridge the aforementioned research gap, we introduce the Chinese Multimodal Metaphor Dataset (MultiCMET). This dataset comprises text-image pairs extracted from Chinese advertisements, encompassing textual slogans and their corresponding image counterparts. We annotate not only the occurrence of metaphor as in previous work, but also how metaphor arises in multimodality. Specifically, we provide explicit reasoning for multimodal metaphors by annotating both source and target domains. The quality control and agreement analyses

for multiple annotators are also presented. Furthermore, we construct a domain lexicon which contains hierarchical domain categories based on the upper/lower relation in WordNet's set of nouns (Miller, 1995) along with corresponding examples. We introduce three evaluation tasks to assess the capabilities of multimodal metaphor understanding: metaphor detection, sentiment analysis, and a novel task, namely domain classification, which focuses on classifying the source and target domains of metaphors—an area that has not been previously explored. We propose the Cascading Domain Knowledge Integration (CDKI) benchmark, which leverages domain-specific features derived from the domain lexicon. During the evaluation phase, we employ multiple benchmarks with CDKI to assess the performance of metaphor understanding tasks, showcasing the effectiveness of CDKI. Our contributions are as follows:

- Creation of a Chinese dataset, MultiCMET, consisting of 13,820 text-image pairs from advertisements.

- The domain classification approach and the construction of the domain lexicon provide a deeper understanding of metaphor usage and facilitate further research.

- We propose three tasks to evaluate multimodal metaphor understanding capabilities, including a new task: domain classification. The CDKI benchmark demonstrates the capabilities of the MultiCMET dataset and its usefulness in advancing multimodal metaphor understanding.

## 2 Related Work

### 2.1 Multimodal Metaphor Datasets

Research on multimodal metaphor datasets has just started, and few multimodal metaphor datasets have been created, although a number of textual metaphor datasets have been introduced for metaphor processing in NLP (Steen et al., 2010; Birke and Sarkar, 2006; Mohammad et al., 2016). Shutova et al. (2016) and Zhang et al. (2021) are among the very few to construct multimodal samples to explore multimodal metaphor processing. However, both types of dataset suffer from the same two deficiencies: (1) they only contain English samples, and (2) they focus on metaphor detection (metaphorical or literal), and they have not

achieved a profound understanding of multimodal metaphor or how metaphor arises. This makes our dataset, which contains Chinese samples with annotations of mechanisms for metaphor development, different from existing datasets.

## 2.2 Metaphor Understanding

Certain tasks must be completed before decoding metaphorical messages automatically. We propose three tasks to evaluate multimodal metaphor understanding capabilities, namely metaphor detection, domain classification, and sentiment analysis. Previous research has focused on a variety of approaches to metaphor processing in textual data. Early studies of metaphor focused on the techniques of machine learning and hand-constructed knowledge (Mason, 2004). Others used distributional clustering (Shutova et al., 2013) or unsupervised approaches (Shutova et al., 2017; Mao et al., 2018). Recently, others explored deep learning models to illuminate metaphor (Kuo and Carpuat, 2020; Liu et al., 2020; Choi et al., 2021; Ge et al., 2022). Several recent researchers have become interested in multimodal metaphor detection, such as (Shutova et al., 2016; Kehat and Pustejovsky, 2020; Su et al., 2021), who have explored the fusion of textual and image modalities to detect metaphor. Their results demonstrated the positive effect of combining textual and image features for metaphor detection.

However, they only extracted features from English samples without considering cultural and language variations. Moreover, apart from multimodal metaphor detection, the tasks in our work related to a deep understanding of metaphor and explicit reasoning of how multimodal metaphor arises, like target/source domain classification, which has rarely been studied before.

## 2.3 Domain Categories in Metaphors

The source domain refers to a familiar concept or domain that is used to understand or describe another concept or domain, known as the target domain (Lakoff and Johnson, 1980). Arriving at a full understanding of metaphorical messages requires identifying two distinct domains that contribute to the occurrence of metaphor. One notable and influential example is the Master Metaphor List (Lakoff, 1994), which includes a wide range of source and target domain categories. This list provides examples of how concepts from the source domain are metaphorically mapped onto the target domain.

For instance, the source domain includes categories such as physical object, living being, location, container, path, gateway, depth, motion, journey etc. The target domain includes categories such as life, death, time, past, progress, mind, ideas, problem, and others. The source and target domain categories from the Master Metaphor List have been widely adopted in NLP research (Barnden and Lee, 2007; Lönneker, 2004) and have been extended with novel categories in subsequent work (Shutova and Teufel, 2010).

However, the domain categories of previous studies are often created based on intuition and existing metaphor examples, which means that they may not capture the full range of metaphorical expressions across different contexts and domains. Additionally, there is a lack of clear structuring principles in the mapping (Lönneker-Rodman, 2008), resulting in confusion regarding taxonomical levels (Shutova and Teufel, 2010).

Our work in this paper, offers significant advantages compared to previous domain classification approaches. It allows for the categorization of domains in a broad context and provides clear structuring principles. Furthermore, it establishes clear taxonomical levels and distinct classes of domains.

# 3 MultiCMET Dataset

## 3.1 Data collection

MultiCMET aims to provide a Chinese annotated dataset to facilitate the research on automatic multimodal metaphor understanding. We chose advertisements with both text and image as our data source because they provide an important context for multi-modal metaphor research (Forceville, 2017). Whether they are commercial or public service ads, these two most common types of advertisements are particularly attractive as a means of communication and contain a large amount of metaphorical information and visual and language features.

We collected potential Chinese metaphorical ad samples by using Chinese keywords to search on Baidu and Bing, the two most popular search engines in China. In particular, the overall process of Chinese data selection was driven by researchers who are native Chinese speakers. Specifically, we compiled a list of keywords related to "advertisement" and "metaphor" and queried these keywords through search engines. The selected keywords encompassed everyday products such as "mobile",

"car", and "pork"; public service announcement topics such as "smoking", "bullying", and "driving"; as well as widely studied relevant topics to metaphor in pure linguistics such as "anger", "color", and "animal". We also referred to a Master Metaphor List (Lakoff, 1994) to select target/source domains in conceptual metaphor, looking for keywords such as "change", "emotion", "people", "beliefs", etc. The detailed list of keywords can be found in Appendix A.1. Additionally, we collected potential metaphorical ad samples of Chinese metaphors from a large-scale commercial advertising dataset with images and internal texts from the IFlytek Advertising Image Classification Competition released in 2021.[2] We obtained a total of 20,672 images, with 10,325 obtained through keyword searches and 10,347 obtained from the dataset.

After obtaining the noisy images, we eliminated duplicate images by comparing their MD5 encoding(Rivest, 1992). We manually screened out images that were not advertisements, and those that were blurry or smaller than 350 x 350 pixels. In addition, we removed images without any internal text. To extract textual information from the remaining images, we employed the paddle OCR model (Du et al., 2020), which enabled us to process the text separately. However, due to potential inaccuracies in the OCR output, we manually corrected the extracted text to ensure accuracy. After data cleaning, we finally obtained 13,820 image-text pairs.

## 3.2 Annotation model

We annotated the text-image advertising pairs, identifying the metaphors (metaphorical or literal); (if metaphorical) categories of target and source domain (13 categories: relation, communication, attribute, psychological feature, person, plant, animal, process, event, substance, natural object, artifact, or location); and sentiment category(the sentiment metaphors evoke, namely negative, neutral, or positive). The annotation model was Metaphor-Model=(Occurrence, TargetCategory, SourceCategory, SentimentCategory). An example of the annotation can be seen in Figure 2.

## 3.3 Data annotation

**Metaphorcial or literal** We followed the approach of metaphor identification used in Multi-MET (Zhang et al., 2021)to identify data. The

[2]https://aistudio.baidu.com/aistudio/datasetdetail/102279

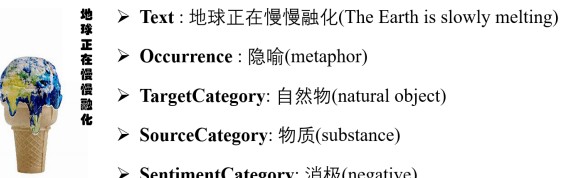

> ➤ **Text**：地球正在慢慢融化(The Earth is slowly melting)
> ➤ **Occurrence**：隐喻(metaphor)
> ➤ **TargetCategory**: 自然物(natural object)
> ➤ **SourceCategory**: 物质(substance)
> ➤ **SentimentCategory**: 消极(negative)

Figure 2: An example of a metaphorical annotation

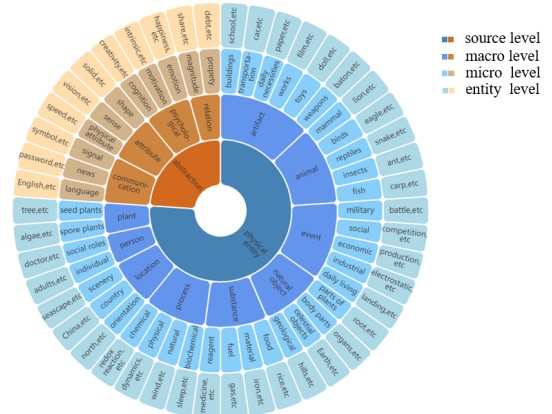

Figure 3: The Lexical Hierarchy of the Domain. Each layer, ranging from the innermost to the outermost, signifies a progressively refined level of granularity.

metaphor annotation was conducted at the relational level, which involved identifying metaphorical relations between expressions from the source and target domains. In this process, annotations of text modality were made based on language clues, while annotations of image modality were made based on visual features. Both methods determined the linguistic categories of text and images. Specifically, the annotators identified metaphorical text and image pairs by scrutinizing the incongruous units and explaining a non-reversible "*A is B*" identity relation, where two domains were expressed by multiple modalities.

**Domain categories and domain lexicon** Metaphor involves using one concrete concept to comprehend and express another abstract concept, aiding reasoning and communication (Lakoff and Johnson, 1980). As a starting point for our hierarchical domain classifications, we use "physical entity" to refer to a concrete and tangible object or substance, and "abstraction" to refer to a concept or idea representing a general quality, attribute, or state. These two categories form the "source level" of our classification.

To further organize the domain categories, we refer to the hierarchical structure of WordNet (Miller, 1995) and divide the domains into three additional

| | Total Samples | Metaphorical Samples | Literal Samples | Total Words | Avg Words |
|---|---|---|---|---|---|
| Train | 9,674 | 4,414 | 5,260 | 270,872 | 28 |
| Val | 1,382 | 630 | 751 | 46,249 | 33 |
| Test | 2,764 | 1,262 | 1,503 | 82,920 | 30 |
| Total | 13,820 | 6,306 | 7,514 | 400,041 | 29 |

Table 1: An overview of the statistics of the dataset

hierarchical levels: macro level, micro level, and entity level (as shown in Figure 3). At the macro level, the source level is subdivided into 13 categories. This includes 9 physical entity categories such as person, animal, artifact, event, location, and process, as well as 4 abstract categories including relation, communication, attribute, and psychological. The division at this level is based on the upper/lower relations in WordNet's set of nouns. Due to some overlap among the original 25 categories in WordNet, we made adjustments and reclassified the domains into these 13 classes. For example, in Chinese, the meanings of person and people are similar, so they are not strictly distinguished. Similarly, feelings, motivations, and cognition share common psychological features and are classified as subcategories of psychological features. Food, being part of substance, is classified as a subclass of substance. The micro level comprises category words obtained by further subdividing the macro level categories. For example, animals are divided into subcategories such as mammals, birds, reptiles, insects, and fish. The entity level represents specific examples within the micro level. These examples are representative of the categories. For instance, "tiger", "mouse", and "bat" are specific examples of mammals at the entity level.

We used the 13 categories from the macro level of domain classifications to annotate the source and target domain categories of metaphors. We didn't use the 46 categories from the micro level because they were more complex and detailed. The macro level categories offered a more manageable and meaningful level of abstraction for our annotation task. Furthermore, using the domain classifications described above, we construct a domain-specific lexicon that includes vocabulary corresponding to each domain category, along with exemplar words. The lexicon consists of 2,755 words, including 61 category words and 2,694 specific example words. Refer to Appendix A.5 for a detailed domain lexicon.

**Sentiment categories** The comprehension of metaphor primarily entails identifying the mapping

of two domains and extracting properties conveyed by metaphors. One essential aspect of metaphor is sentiment, which has been found to have a stronger emotional impact than literal expressions in previous studies (Citron and Goldberg, 2014; Mohammad et al., 2016). Therefore, to investigate whether the sentiment impact of metaphors is more pronounced than that of literary expressions from multicultural and multimodal perspectives, we annotated sentiment in our dataset. The sentiment was categorized into negative, neutral, and positive.

### 3.4 Annotation process and quality control

We employed an expert-based approach to annotate data for three challenging metaphor understanding tasks, with annotations completed by five Chinese native speakers acting as annotators. The five annotators were divided into three small groups, with two groups consisting of two members each, and the third group consisting of only one member. In cases where the two-member groups could not reach a consensus, the single-member group participated in the final decision. If there were no disagreements between group members, the annotation task was considered complete. Otherwise, the single-member group would re-annotate the data. Finally, if there were disagreements in the annotations from all groups, everyone would discuss and decide on the annotation to ensure its accuracy and consistency. For samples with discrepant annotations, we held multiple in-group discussions and conducted checks and modifications to improve annotation consistency and accuracy.

To improve annotation quality, we took several effective measures. We established strict criteria and documentation for every annotation option, including detailed explanations, extensive examples, and notes. For the convenience of annotation, we provide an interface for the annotators as detailed in Appendix A.2. Additionally, prior to each annotation session, we held a training course to provide guidance. During the pre-annotation process, we adjusted the course and guidance documents to address any issues, ensuring that the annotation document was comprehensive and definitive before large-scale annotation.

Kappa score (Fleiss, 1971) was used to measure classification consistency, with scores of $\varkappa$ =0.69, $\varkappa$ =0.66, $\varkappa$ =0.62, and $\varkappa$ =0.77 achieved for identifying metaphors, identifying target domain categories, identifying source domain categories, and

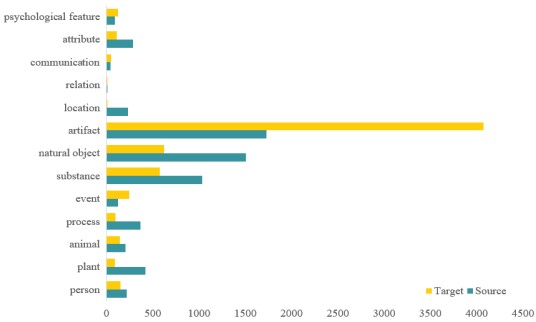

Figure 4: Distribution of Categories in the Source and Target Domains

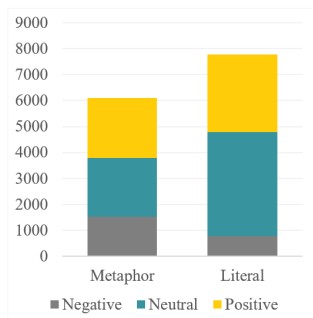

Figure 5: Distribution of Emotional Categories in Metaphorical and Non-Metaphorical Data

identifying emotional categories, respectively, indicating the reliability of our annotations.

## 4 Dataset Analysis

Refer to Table 1 for an overview of the dataset statistics. We conducted an analysis of category distribution in both the source and target domains, as presented in Figure 4. It is evident that the categories "artifact", "natural object" and "substance" appear most frequently in both domains, indicating the preference for these tangible categories in expressing ideas and emotions in metaphorical advertising. Conversely, the categories "communication", "relation" and "psychological" have relatively lower frequencies, as they often involve more abstract and intricate concepts that require a deeper understanding of contextual and background knowledge. Furthermore, among the 13 categories examined, the source domain exhibits a more dispersed distribution, suggesting a diverse utilization and manifestation of source domains, while the target domain revolves around more centralized themes and situations compared to the source domain.

Figure 5 demonstrates varying emotional distributions across different metaphors. In metaphorical data, positive and neutral emotions are evenly distributed, while negative emotions account for a smaller portion. This indicates that metaphorical expressions in advertising often correlate with positive or neutral attitudes. In non-metaphorical data, neutral emotions dominate the dataset, followed by positive emotions, with negative emotions being the least prevalent. This reflects the directness and clarity of non-metaphorical expressions, which are typically utilized to convey widely accepted information or depict everyday life scenarios. By comparing the statistical data of metaphorical and non-metaphorical expressions, we can observe that the proportion of neutral data in metaphorical expressions is lower than that in non-metaphorical expressions, suggesting that metaphorical expressions tend to encompass a richer emotional range. Our findings are consistent with previous research suggesting that metaphors convey more emotions or feelings than simple prose (Citron and Goldberg, 2014).

## 5 Methodology

Taking into account the significance of domain knowledge in metaphor detection, we propose a benchmark called Cascading Domain Knowledge Integration (CDKI) that aims to enhance the model's ability to detect metaphors by incorporating domain knowledge. The schematic diagram of CDKI is illustrated in Figure 6.

Introducing domain knowledge into models poses a challenge due to inherent differences in how it is presented in text and images. Textual domain knowledge is typically easier to extract, expressed explicitly through specific vocabulary and sentence structures. Conversely, domain knowledge in images tends to be more implicit. In the following section, we will elaborate on how CDKI addresses this disparity and achieves the extraction and integration of domain knowledge.

Regarding the text modality, we employ a segmentation technique to preserve multiple inherent domain-specific vocabularies in the textual data by extracting nouns. These nouns are then connected using the [SEP] token to form $K_{text}$, which serves as the domain knowledge for the text modality.

Regarding the image modality, we have constructed a cascaded domain word set to introduce and enhance the dependency relationships of domain knowledge within the images. Specifically,

the cascaded domain word set for an image consists of three parts: the macro-level domain word set, the micro-level domain word set, and the entity-level domain word set. These word sets are composed of vocabulary corresponding to the respective levels in the domain lexicon. Taking the macro-level domain word set as an example, we will describe the construction method of the cascaded domain word set for images. By utilizing Clip (Radford et al., 2021), we obtain the probabilities of each vocabulary in the macro-level of the domain dictionary appearing in the image. We then retain the top $n$ vocabularies with the highest probabilities as the macro-domain word set $Set_{macro}$ for that particular image. The construction method for the micro-domain word set, $Set_{micro}$, and the entity-domain word set, $Set_{entity}$, follows the same approach. Each set consists of p and q vocabularies, respectively. Collectively, these three sets constitute the domain knowledge within the image, denoted as $K_{image}$:

$$K_{image} = Set_{macro} + Set_{micro} + Set_{entity} \quad (1)$$

Lastly, the domain knowledge from the image and text modalities are combined to generate the final domain knowledge, denoted as $K_{pair}$, for the image-text pairs:

$$K_{pair} = K_{image} + K_{text} \quad (2)$$

In addition, we conducted multiple sets of experiments to investigate the impact of different vocabulary sizes on the model results while keeping the text attributes fixed. The details of these experiments can be found in Appendix **??**. Ultimately, we determined that the optimal values for $n$, $p$, and $q$ are 2, 3, and 5, respectively.

To incorporate the domain knowledge $K_{pair}$ into the model, we employ BERT (Devlin et al., 2018) to convert the $K_{pair}$ into vectors, which serve as the domain feature input for the image-text pairs. We utilized BERT and ResNet50 (He et al., 2016) to obtain textual and visual features respectively. Subsequently, we utilize cross-attention mechanism (Vaswani et al., 2017) to jointly model the domain features with the image and text features. This approach enhances the interaction of domain knowledge between different features. The resulting outputs are concatenated and fed into the softmax function to obtain the final result.

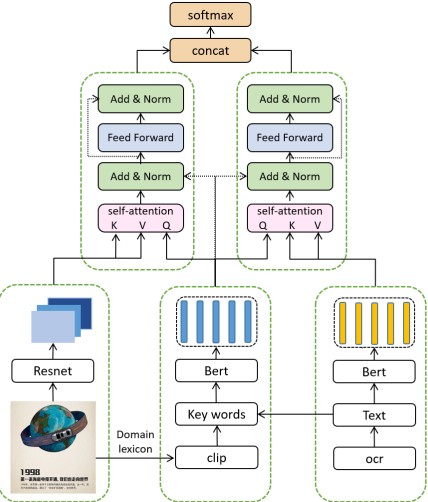

Figure 6: Cascading Domain Knowledge Integratio (CDKI) illustration

# 6 Experiments

## 6.1 Baselines

In this section, we introduce the baselines used in our experiments. Specifically, we adopted random, text-based, image-based, and multi-modal models as the baseline models and compared their performance with our proposed model. Due to the fact that metaphors, irony, and hate speech fall under the category of figurative language, there exists a certain degree of similarity between these three tasks(Abulaish et al., 2020). Thus, we selected these multi-modal methods as our control models.

**Random**: It denotes random predictions based on the data.

**Resnet50** (He et al., 2016): It is a 50-layer convolutional neural network with residual connections. Image features are extracted from its last convolutional layer's output vector.

**VGG16** (Simonyan and Zisserman, 2014): It enhances network depth and the expression capability of non-linear features by stacking consecutive 3x3 convolutional layers and max pooling layers.

**Bi-LSTM** (Hochreiter and Schmidhuber, 1997): It's a bidirectional recurrent neural network that combines the outputs of forward and backward LSTM to capture global context in a sequence.

**Bert** (Devlin et al., 2018) and **Roberta** (Liu et al., 2019): We conducted the experiment using two widely-used Chinese transformer-based pre-trained language models, namely bert-based-chinese and roberta-base-chinese, as baselines.

**Res-BERT** (Zhang et al., 2021): It uses

| | | Metaphor | | Target domain | | Source domain | | Sentiment | |
|---|---|---|---|---|---|---|---|---|---|
| Type | Model | ACC | F1 | ACC | F1 | ACC | F1 | ACC | F1 |
| Random | - | 0.4997 | 0.3332 | 0.0755 | 0.0108 | 0.0748 | 0.0107 | 0.3391 | 0.1688 |
| Text | Bi-LSTM | 0.6455 | 0.5141 | 0.3158 | 0.1164 | 0.3265 | 0.1186 | 0.5329 | 0.5257 |
| | BERT | 0.6774 | 0.6429 | 0.3459 | 0.1077 | 0.3681 | 0.1254 | 0.5784 | 0.5457 |
| | Roberta | 0.6749 | 0.6578 | 0.3564 | 0.1038 | 0.3645 | 0.1367 | 0.5836 | 0.5531 |
| Image | VGG16 | 0.6355 | 0.5022 | 0.3214 | 0.1030 | 0.3233 | 0.1297 | 0.5426 | 0.5361 |
| | ResNet50 | 0.6624 | 0.6312 | 0.3347 | 0.1123 | 0.3476 | 0.1447 | 0.5579 | 0.5458 |
| Text+Image | Res-BERT | 0.7002 | 0.6745 | 0.3680 | 0.1396 | 0.3868 | 0.1352 | 0.6048 | 0.5727 |
| | VilBERT | 0.7231 | 0.6718 | 0.3683 | 0.1224 | 0.3754 | 0.1509 | 0.6192 | 0.5801 |
| | VisualBERT | 0.7277 | 0.6899 | 0.3783 | 0.1352 | 0.4041 | 0.1594 | 0.6231 | 0.6055 |
| | HFM | 0.7363 | 0.6757 | 0.3828 | 0.1355 | 0.3924 | 0.1656 | 0.6112 | 0.5802 |
| | CES | 0.7430 | 0.6996 | 0.3952 | 0.1421 | 0.4339 | 0.1845 | 0.6303 | 0.5986 |
| | Our model | **0.7632** | **0.7289** | **0.4377** | **0.1754** | **0.4692** | **0.1962** | **0.6693** | **0.6337** |

Table 2: Results on proposed dataset (MultiCMET). Best in bold.

ResNet50 to extract image features and BERT to encode text. Then, the two sets of features are concatenated to form a fused feature vector, which is input into a fully connected layer for classification.

**VilBERT** (Lu et al., 2019) and **VisualBERT** (Li et al., 2019): They are both multimodal language models based on the Transformer architecture. VilBERT utilizes a dual-stream attention mechanism to encode visual and textual information. VisualBERT employs a self-attention mechanism to encode images and text in a single stream manner.

**HFM** (Cai et al., 2019): The multimodal fusion model combines image, attribute, and text features using Bi-LSTM and MLP to generate a unified representation for ironic prediction tasks.

**CES** (Blaier et al., 2021): This method improves hate speech meme classifier accuracy by leveraging image-caption to enhance both multimodal and unimodal models.

### 6.2 Implementation

Our model was constructed using the PyTorch framework (Paszke et al., 2019), and we pre-processed our data utilizing the pandas and NumPy libraries. Our optimization algorithm of choice was Adam (Kingma and Ba, 2014), with cross-entropy serving as our loss function. After fine-tuning the baseline, we selected the best-performing model and hyperparameters to be tested on the reserved testing set. Table 3 presents a comprehensive overview of the key hyperparameters employed in our experimental study.

### 6.3 Results and discussion

We evaluate our model's performance by comparing it with baseline models using accuracy and F1

| Hyper-parameters | Value |
|---|---|
| Image embedding size | 2048 |
| Dropout | 0.3 |
| Text max length | 64 |
| Batch size | 32 |
| Learning rate | 3e-5~1e-4 |
| Modality fusion size | 512 |

Table 3: Hyperparameters.

scores, as shown in Table 2. Our model excels in metaphor detection, followed by metaphor sentiment detection, and performs weakest in target and source domain detection. The subpar performance in target and source domain detection can be attributed to two factors. Firstly, with 13 categories, there is an excessive number of category features that the model struggles to accurately capture. Secondly, an issue of data imbalance exists among different categories, resulting in inadequate generalization capabilities of the model for minority categories. Nonetheless, our model outperforms all baseline models, validating its effectiveness.

From the table, it is evident that models solely relying on image features exhibited subpar performance. Among them, ResNet50 outperformed VGG16 due to its utilization of residual connections to mitigate the issue of gradient vanishing.

Text-based methods exhibited superior performance compared to image-based methods, as text can convey more contextual information and provide valuable context to the model. Both Bert and Roberta, having been fully trained on large-scale language corpora, outperformed Bi-LSTM.

The multimodal approach yielded the best results, as the fusion of image and text features enhanced the detection performance of the model.

Our model achieved the highest performance in the multimodal setting, In comparison to the CES model that incorporates image captions as additional features, the inclusion of domain knowledge provides the model with richer and more valuable information. While HFM utilizes image attributes as additional features, our approach introduces domain attributes with hierarchical relationships, enabling the model to leverage higher-level concepts and semantic associations. This incorporation of hierarchical domain attributes effectively enhances the performance of the model.

We also performed a qualitative analysis of the wrongly classified samples. We examined approximately 50 misclassified instances and categorized the reasons for the errors, which are detailed in Appendix A.4.

## 7 Conclusion

In conclusion, this paper addresses the limitations in research on multimodal metaphors beyond the English language and the lack of exploration in categorizing source and target domains in metaphors. It introduces the MultiCMET dataset, a large-scale multimodal Chinese metaphor dataset consisting of text-image pairs from advertisements. The dataset includes manual annotations of metaphors, domain categories, and conveyed sentiments. Additionally, a domain lexicon is constructed, providing hierarchical domain classifications and corresponding examples. We propose the CDKI benchmark, which leverages domain-specific features for better metaphor understanding.

This paper significantly contributes to the field of multimodal metaphor understanding by providing a comprehensive dataset, novel methodologies, and benchmark results. It opens up avenues for future research in exploring metaphors across different languages, enhancing metaphor comprehension, and addressing the cognitive mechanisms involved in metaphor processing. The availability of the dataset and code further encourages the research community to expand and improve upon the findings presented in this work.

## Limitations

One limitation of the current version of Multi-CMET is its exclusive coverage of the Chinese language. It is highly necessary to develop the MultiCMET dataset for other languages in order to facilitate comparative research on multimodal metaphors between different language systems. This expansion would help uncover biases present in current metaphor models and provide a deeper understanding of the cognitive mechanisms underlying metaphor comprehension. Moreover, a dataset encompassing more diverse languages and cultures would greatly benefit the models' capacity to comprehend multimodal metaphors. We encourage researchers to embrace the challenging yet captivating task of expanding MultiCMET by incorporating data from additional languages in future endeavors.

## Ethics Statement

Based on the source of the data, our MultiCMET dataset can be divided into two parts. The first part consists of data obtained through querying search engines, specifically Baidu and Bing. For this portion, we have followed the data use agreements of these search engine querying platforms. We have made our annotations publicly available for this data, and we provide a detailed description of the query process.The second part of the dataset comprises data collected from publicly available datasets that allow data usage for non-commercial academic purposes. We strictly adhere to the terms and conditions of these publicly available datasets. We have released the unique IDs associated with this data, along with our annotations. By ensuring transparency and compliance with data use agreements, we aim to promote responsible and ethical data usage while enabling researchers to access and utilize the MultiCMM dataset effectively.

## Acknowledgements

This work is supported by NSFC Programs (No.62076051).

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

# A Appendix

## A.1 Keyword query

The keywords used during the data collection process are presented in Table 4.

| Topic | Keywords |
|---|---|
| Commercial | mobile phone, tablet, television, laptop computer, pork, beef, lamb, car, motorcycle, bicycle, electric vehicle, tire, engine oil, toothpaste, mouthwash, hair gel, detergent, laundry detergent, moisturizer, sunscreen, shower gel, soap, baby food, stroller, cookware, kitchen utensils, home decor, bedding, snacks, beverages, health supplements, nutritional supplements |
| Public service | smoking, drug prohibition, bullying, campus violence, drunk driving, environmental protection, energy conservation and emission reduction, waste sorting, animal protection, afforestation, reducing plastic pollution, traffic safety, fire safety, cybersecurity, protection of intellectual property rights, anti-fraud, counter-terrorism and riot prevention, drowning prevention, natural disaster response, food safety |
| Metaphor | anger, losing temper, calmness, composure, animal, cat, dog, belief, values, philosophy, happiness, contentment, understanding, change, alteration, color, emotion, red, orange, yellow, people, organization, ethnic group |

Table 4: Key words of each topic.

## A.2 Annotation Interface

Figure 7 presents several examples of our annotated illustrations.

## A.3 Number of vocabulary terms

In this section, we aim to explore the relationship between the number of subset vocabulary in image cascading and model performance. Firstly, we use a baseline model without domain knowledge, where the values of n, p, and q are all 0. The accuracy of this model is 0.7134. Next, we gradually increase the values of n, p, and q while keeping other features constant to analyze their impact on model performance and eliminate interference. The ranges of n, p, and q are [0-5], [0-7], and [0-10] respectively. We select representative data from these ranges and present them in Table 5.

When n, p, and q are all set to 1, the model's accuracy in metaphor recognition is already higher than that of the baseline model without domain knowledge, reaching 0.7272. This indicates that

| n | p | q | ACC |
|---|---|---|---|
| 0 | 0 | 0 | 0.7134 |
| 1 | 1 | 1 | 0.7272 |
| 1 | 2 | 4 | 0.7396 |
| 1 | 3 | 5 | 0.7452 |
| 1 | 6 | 9 | 0.7426 |
| 1 | 7 | 10 | 0.7315 |
| 2 | 2 | 2 | 0.7445 |
| 2 | 2 | 4 | 0.7538 |
| 2 | 3 | 5 | **0.7632** |
| 2 | 6 | 9 | 0.7499 |
| 2 | 7 | 10 | 0.7433 |
| 3 | 3 | 3 | 0.7516 |
| 3 | 2 | 4 | 0.7501 |
| 3 | 3 | 5 | 0.7541 |
| 3 | 6 | 9 | 0.7432 |
| 3 | 7 | 10 | 0.7389 |

Table 5: Model accuracy with different values of $n$, $p$ and $q$. The values of $n$, $p$, and $q$ represent the respective quantities of vocabulary terms in $Set_{macro}$, $Set_{micro}$, $Set_{entity}$.

introducing domain knowledge is helpful in improving model accuracy. As the values of n, p, and q increase, the accuracy of the model in metaphor recognition also improves. When n = p = q = 3, the accuracy of the model is 0.7516, higher than the model's performance when n = p = q = 1. This suggests that increasing the number of vocabulary can improve the model's accuracy.

To analyze the relationship between subset vocabulary and model accuracy, we conducted experiments using controlled variables. Results showed that increasing values of p and q improved model accuracy, while keeping n constant. Micro-domain and entity-specific vocabulary helped capture detailed features and enhance recognition accuracy. However, higher values of n, p, and q were not always better. The parameters n = 2, p = 3, and q = 5 achieved optimal results, while n = 3, p = 6, and q = 9 did not perform as well despite having a larger vocabulary. Excessively high values of n, p, and q introduced noise, resulting in decreased accuracy.

In conclusion, incorporating domain knowledge can enhance the model's recognition accuracy, and expanding the subset vocabulary in image cascading can improve model performance. However, an excessive number of vocabulary may introduce

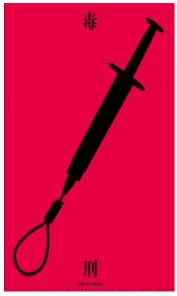

➤ **Text**: poison punishment

➤ **Occurrence**: metaphor

➤ **Target category**: artifact

➤ **Source category**: artifact

➤ **Sentiment category**: negative

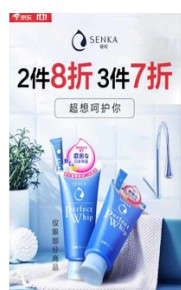

➤ **Text**: buy 2 get 20% off, buy 3 get 30% off.

➤ **Occurrence**: non-metaphor

➤ **Sentiment category**: neutral

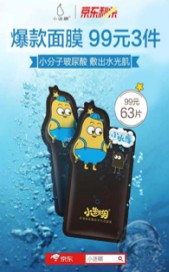

➤ **Text**: best-selling facial mask

➤ **Occurrence**: metaphor

➤ **Target category**: artifact

➤ **Source category**: substance

➤ **Sentiment category**: positive

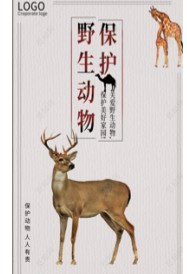

➤ **Text**: protecting wildlife

➤ **Occurrence**: non-metaphor

➤ **Sentiment category**: positive

Figure 7: Examples of annotations.

noise and lead to performance deterioration. This is further supported by the parameters n = 2, p = 7, q = 10, and n = 3, p = 7, q = 10. Hence, we select the parameters n = 2, p = 3, and q = 5 as the subset vocabulary size, which yielded the best experimental outcomes.

### A.4 Error analysis

One type of error is when the model misclassifies metaphoric text and images as non-metaphoric. This often occurs when the target or source domain in the image is heavily distorted or partially visible, limiting the model's ability to learn domain information. As a result, conflicts between the text and image may go undetected, leading to the incorrect identification of metaphors. For instance, in Figure 8 on the left, the obscured tiger tail prevents proper recognition of the conflicting elements with the euro symbol, resulting in the image being mistakenly classified as non-metaphoric.

The second classification error occurs when the model misclassifies non-literal text as metaphorical. This can happen due to changes in artistic style or an information overload, making it difficult for the model to learn domain-specific information and detect conflicts between domains. In the Figure 8 on the right, the text itself lacks metaphorical meaning, but the model mistakenly identifies it as metaphorical. Additionally, the model wrongly

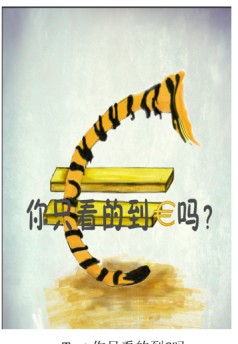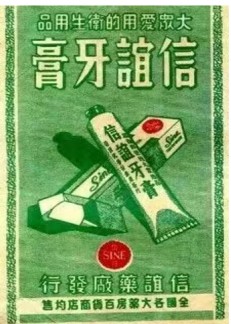

Text:你只看的到€吗
(Do you only see €?)

Grounded：Metaphor
Predict：non-Metaphor

Text: 信谊牙膏
(Xinyi Toothpaste)

Grounded：non-Metaphor
Predict：Metaphor

Figure 8: Two examples of identification errors.

categorizes the target domain as "artifacts" and the source domain as "natural objects." Analysis suggests this may be because the image is hand-drawn, leading the model to incorrectly interpret certain elements as natural objects and perceive a conflict between toothpaste and natural objects. As a result, the text-image pair is mistakenly classified as a metaphor.

### A.5 Domain Lexicon

| | | | |
|---|---|---|---|
| physical entity | person | individual | adults,children,teens,etc |
| | | social roles | doctor,engineer,father,etc |
| | animal | mammal | tiger,mouse,bat,etc |
| | | birds | eagle, peacock, penguin, etc |
| | | reptiles | snake, turtle, lizard, etc. |
| | | insects | butterfly, bee, ant, etc. |
| | | fish | goldfish, carp, grass carp, etc. |
| | plant | spore plants | algae, seaweed, moss, etc. |
| | | seed plants | tree, grass, flower, etc. |
| | artifact | buildings | library, school, office building, etc. |
| | | transportation | car, boat, train, etc. |
| | | daily necessities | scissors, paper, pen, etc. |
| | | works | film, book, poem, etc. |
| | | toys | balloon, doll, building blocks, etc. |
| | | weapons | baton, nunchaku, firearm, etc. |
| | natural object | parts of plants | root, stem, leaf, etc. |
| | | body parts | organs, lungs, heart, etc. |
| | | celestial objects | Earth, Moon, stars, etc. |
| | | geological structures | hills, plateaus, mountain peaks, etc. |
| | substance | food | beverage, rice, wine, etc. |
| | | material | iron, water, wood, etc. |
| | | fuel | gas, oil, methane hydrate, etc. |
| | | reagent | medicine, cigarette, bleach, etc. |
| | event | military activities | battle, siege, desertion, etc. |
| | | social activities | competition, performance, etc. |
| | | economic activities | production, consumption, etc. |
| | | industrial activities | electrostatic precipitation, distillation, hydrolysis, etc. |
| | | activities of daily living | landing, recovery, wearing, etc. |
| | location | orientation | north, south, west, etc. |
| | | country | China, America, Germany, etc. |
| | | scenery | seascape, mountain view, etc. |
| | process | biochemical processes | metabolism, respiration, sleep, etc. |
| | | natural processes | wind, rain, snow, etc. |
| | | physical processes | light propagation, dynamics, heat conduction, etc. |
| | | chemical process | redox reaction, reaction, metal corrosion, etc. |
| abstraction | relation | magnitude relationship | share, ratio, quantity, etc. |
| | | property relationship | economic situation, debt, etc. |
| | communication | language | English, Chinese, French, etc. |
| | | news | password, email, news, etc. |
| | | signal | phone number, symbol, etc. |
| | attribute | physical attributes | temperature, speed, density, etc. |
| | | sense | vision, taste, touch, etc. |
| | | shape | solid, linear, triangle, etc. |
| | psychological | cognition | creativity, thinking, concept, etc. |
| | | motivation | intrinsic motivation, etc. |
| | | emotion | happiness, pride, sadness, etc. |

Table 6: Domain Lexicon.