# OpenReview forum: "MultiCMET: A Novel Chinese Benchmark for Understanding Multimodal Metaphor"
_EMNLP/2023/Conference — EMNLP 2023 Findings_

### Official Review · Reviewer_MYGj · 2023-07-26

**Typos Grammar Style And Presentation Improvements:** line 59，290，318
**Soundness:** 3

**Excitement:**

4: Strong: This paper deepens the understanding of some phenomenon or lowers the barriers to an existing research direction.

**Paper Topic And Main Contributions:**

This paper proposed a multimodal Chinese metaphor dataset, MultiCMET, consisting of 13,820 text-image pairs of advertisements with manual annotations of the occurrence of metaphors, source and target domain categories, and sentiments metaphors convey. Furthermore, it developed a domain lexicon that encompasses categorizations of domains and corresponding examples. Finally, it propose the Cascading Domain Knowledge Integration (CDKI) benchmark, which utilizes domain-specific features from the domain lexicon to enhance the understanding of metaphors.

**Questions For The Authors:**

As mentioned in weakness, what is the significance of metaphor domain classification for metaphor recognition, Does it help to recognize metaphors?

**Reasons To Accept:**

1. The paper is well-written and easy to understand.

2. This work claims to firstly propose a publicly available large-scale Chinese metaphor dataset, and will publish the data and code. Although some Chinese metaphor datasets existed in the past, the poor annotation quality of these datasets makes them difficult to use. Therefore, the new Chinese metaphor dataset proposal in this work is significant to the community.


**Reasons To Reject:**

A contribution of this paper is the metaphor domain classification, which takes identifying the source and target domains of metaphors as a new task. For example, in “The earth is slowly melting”, the target category for metaphor is natural object, and the source category is substance. However, I'm not sure about the significance of identifying source categories and target categories for metaphors. Does it help to identify metaphors better? But no conclusions can be drawn from the experimental results. Perhaps the authors could elaborate more on it or demonstrate in experiments that it helps metaphor.

**Reproducibility:**

5: Could easily reproduce the results.

**Reviewer Confidence:**

3: Pretty sure, but there's a chance I missed something. Although I have a good feel for this area in general, I did not carefully check the paper's details, e.g., the math, experimental design, or novelty.

---

> ### Author Rebuttal · Authors · 2023-08-28
>
> Concerning the Question, the classification of metaphorical domains contributes to the identification of metaphors. Metaphor involves mapping concepts from a source domain to a target domain, where both domains and their mapping relationship constitute a metaphor. For instance, as shown in Figure 2 of this study, the concept of ice cream (source domain) is mapped onto the earth (target domain) to convey the notion of global warming. The purpose of classifying metaphorical domains is to gain a better understanding of the mapping relationships between these domains. The key aspect of metaphor identification lies in capturing the incongruity of domain mappings. We construct a domain dictionary based on the classification results of metaphorical domains, providing qualitative descriptions of the mapping relationships between domains. In Table 2 of the paper, our model demonstrates a significant improvement in metaphor recognition compared to other models that do not utilize the domain dictionary. This indicates that through the use of the domain dictionary, the model is better able to capture the mapping relationship between the source and target domains.

---

### Official Review · Reviewer_Szae · 2023-08-05

**Soundness:** 4

**Excitement:**

3: Ambivalent: It has merits (e.g., it reports state-of-the-art results, the idea is nice), but there are key weaknesses (e.g., it describes incremental work), and it can significantly benefit from another round of revision. However, I won't object to accepting it if my co-reviewers champion it.

**Paper Topic And Main Contributions:**

This paper presents research on multimodal metaphor understanding in the Chinese language.  It makes several contributions towards  multimodal metaphor understanding.

The paper introduces MultiCMET, a Chinese dataset consisting of 13,820 text-image pairs from advertisements.

The paper proposes a domain classification approach and lexicon construction.

The paper introduced a new task to evaluate the capabilities of multimodal metaphor understanding models.

**Questions For The Authors:**

The cultural variation in representing drunk using animals and "泥" (mud) a great approach. Have you considered using some annotators who are not native Chinese speakers? If so, were there any objective reasons for not selecting them?

**Reasons To Accept:**

Taking an advertising perspective, creating a large-scale multimodal dataset is a clever and efficient approach compared to other sources like literary works, songs, or artists' paintings. Extracting data from creative advertisements and artistic creations offers valuable insights.

Using the domain classification approach further enhances our understanding of metaphorical expressions in different forms of data, mirroring human thinking patterns for problem-solving. For instance, using animals to represent intoxicated individuals or using "泥" (mud) to signify someone heavily drunk in Chinese culture. Such differences can be applied for interdisciplinary analysis, exploring cultural variances resulting from the distribution of animals, and aiding language learners in memorizing visual representations of expressions.

Building a high-quality dataset and lexicon construction require substantial effort and resources.

**Reasons To Reject:**

In the appendix, I noticed some commercial advertisements, which could lower the dataset's overall quality. I didn't find relevant statistical data.

Incorporating artwork and portfolios from art and advertising students could improve the dataset's quality.

**Reproducibility:**

4: Could mostly reproduce the results, but there may be some variation because of sample variance or minor variations in their interpretation of the protocol or method.

**Reviewer Confidence:**

3: Pretty sure, but there's a chance I missed something. Although I have a good feel for this area in general, I did not carefully check the paper's details, e.g., the math, experimental design, or novelty.

**Typos Grammar Style And Presentation Improvements:**

In some sections, there is only Chinese without English translation; please provide bilingual translations wherever possible.

---

> ### Author Rebuttal · Authors · 2023-08-28
>
> Concerning the Question, we did not consider using annotators who are non-native Chinese speakers because the metaphor annotation task requires annotators to have an accurate understanding of cultural connotations and the meanings conveyed by vocabulary in different contexts. Non-native Chinese speakers, lacking exposure to the Chinese language and culture, may have insufficient understanding of metaphors and find accurate annotation a challenging task. Therefore, we did not employ non-native Chinese speakers as annotators.

---

### Official Review · Reviewer_RctQ · 2023-08-05

**Soundness:** 3

**Ethical Concerns:**

Yes

**Excitement:**

2: Mediocre: This paper makes marginal contributions (vs non-contemporaneous work), so I would rather not see it in the conference.

**Justification For Ethical Concerns:**

- Although the authors say that they followed the data use agreements of the search engine platforms (Baidu and Bing), I'm not entirely sure if it is fine to make all those advertisement images publicly available.

**Paper Topic And Main Contributions:**

This paper presents a Chinese multimodal metaphor dataset, annotated with metaphors, source and target domains of the metaphors, and conveyed sentiments. It also provides benchmark results for three tasks of (1) metaphor detection, (2) source and target identification, and (3) conveyed sentiment identification.

My detailed feedback is below. Please note that major and minor comments are mixed.
- line 128: that has been previously explored for the text modality
- sec 3.3: did you use the Chinese WordNet? or the English one?
- lines 357-364: it sounds like significant amount of manual revision was performed on the WordNet hierarchy, for addressing both language/cultural gaps and incorrect(?) categories. Do you think the latter issue is due to a deficiency in WordNet in general? or is the revision intended to tailor the categories to this metaphor domain task?
- lines 376-377: isn't it a bit broad? I understand that the broad categories allow manual annotation. However, I'm wondering what kind of applications can be considered with these broad categories. Additionally, I think this doesn't really address the crucial problem of determining the right level of granularity when identifying source and target domains in metaphorical mappings. I was expecting more when I was reading the claim that this paper provides clear structuring principles. Unless the paper provides why these broad categories are better, I think it is an overclaim.
- line 402: since it was mentioned as an expert-based approach, who were the annotators (i.e., backgrounds?) besides that they were native speakers of Chinese? I understand that a training course was provided. I think more detailed descriptions about the annotators could be useful because this is a metaphor annotation task, which is somewhat fuzzy and subjective.
- lines 422-434: It would be good to see some important details of the annotation guidelines because metaphor annotation is super difficult due to its fuzzy and somewhat subjective nature and the dataset is a main contribution of this paper.
- It would be beneficial to discuss disagreements in the annotation process because it will reveal the challenges in the dataset giving a better idea about the tasks.
- Sec 4: frequently mapped domains would be interesting, too.
- lines 525-526: What are n, p, and q?
- Do most images contain text?
- lines 612-613: I didn't quite understand what this sentence meant.
- lines 613-614: Did the model perform relatively well for the categories with more samples?
- lines 644-648: what are some important findings from the qualitative error analysis? They should be in the main text.
- I'm wondering what kind of characteristics these multimodal metaphor images show. For example, many verb metaphors in text show selectional preferences violation. Many adjective+noun metaphors show unusual patterns of word concreteness.
- Many things in the Appendix (Table 3, Figure 6, Figure 7) are in Chinese only. Translations must be provided!

**Questions For The Authors:**

A. It sounds like manual revision was performed on the WordNet hierarchy, for addressing both language/cultural gaps and incorrect(?) categories. Do you think the latter issue is due to a deficiency in WordNet in general? or is the revision intended to tailor the categories to this metaphor domain task? I'm also wondering if you think the proposed categories should be revised for different languages/cultures.

B. I think the annotated categories are quite broad. I understand that the broad categories allow manual annotation. However, I'm wondering why you think these broad categories could be useful.

C. Could you provide more details about the annotators?

D. It would be good to see some important details of the annotation guidelines because metaphor annotation is super difficult due to its fuzzy and somewhat subjective nature and the dataset is a main contribution of this paper.

E. lines 525-526: What are n, p, and q?

F. Do most images contain text?

G. lines 612-613: I didn't quite understand what this sentence meant.

H. lines 613-614: Did the model perform relatively well for the categories with more samples?

I. I'm wondering what kind of characteristics these multimodal metaphor images show, especially in these images for advertisements.

**Reasons To Accept:**

- A multimodal metaphor dataset in non-English could be a valuable resource for future researchers.
- The paper is easy to follow.
- It provides benchmark results, which seem to be from reasonable models.

**Reasons To Reject:**

- I'm not entirely sure how metaphors are defined in the images+texts of this dataset. I cannot understand why some images are metaphorical in the Appendix when only looking at the images. The text descriptions are in Chinese.
- This is a dataset paper, and the annotation guidelines are particularly important for metaphors. Metaphor annotation is notoriously challenging because it is fuzzy and subjective, and multimodal metaphors are even more under-explored. However, the paper doesn't provide sufficient detail about the annotation guidelines and justifications (e.g., how a metaphor is defined and recognized, how sentiment is defined), although the details of the annotation process are provided.
- As a dataset paper, I expect more insightful discussion about the data so that I can determine whether I want to use this dataset or not. For example, what are the common disagreements in the annotations? What kind of characteristics do you observe in the multimodal metaphors? They are all advertisements. Are there any interesting things you can observe with regard to that? This kind of discussion can give a better sense of the dataset and the tasks that can be performed with it.

For these reasons, I think this paper needs more work although I appreciate the work.

**Reproducibility:**

3: Could reproduce the results with some difficulty. The settings of parameters are underspecified or subjectively determined; the training/evaluation data are not widely available.

**Reviewer Confidence:**

4: Quite sure. I tried to check the important points carefully. It's unlikely, though conceivable, that I missed something that should affect my ratings.

**Typos Grammar Style And Presentation Improvements:**

- lines 220-225: source and target domains are not fixed, as you  know. For example, journey could be either source or target, depending on how we use it. I think it would be better to revise this part to avoid misunderstanding about source and target domains. Presenting mapping examples would be better.
- line 341: "source level" <- here, "source" is not related to the source in a metaphorical mapping, right? To avoid confusion, I recommend replacing the word.
- Important/interesting messages from Appendix should be summarized in the main body of the paper.

---

> ### Author Rebuttal · Authors · 2023-08-28
>
> Concerning Question A, we made modifications based on the WordNet(Miller, 1995) structure to adapt it to the context of multimodal metaphor tasks in Chinese, without implying any flaws in WordNet itself. Firstly, we referred to Chinese grammar and made modifications to WordNet. For example, in WordNet, " person " is classified as "person," while " people " is classified as "group." However, in the Chinese context, there is no clear distinction between "person" and "people," and both are categorized as "person." Therefore, we removed the "group" category from WordNet. Secondly, considering the characteristics of metaphor domain tasks, we modified WordNet to better align its categories with metaphor domain classification tasks. For instance, in metaphors, "feelings", "motivations", and "cognition"  are often considered as a single domain. Thus, we classified them as subcategories of psychological features(Lakoff, 1994). In my opinion, the proposed domain categories in the paper are comprehensive, and they can also be applicable to metaphor-related tasks in other languages and cultural contexts. However, it should be noted that the expression of metaphors may vary across different languages and cultural backgrounds. Therefore, I recommend making subtle modifications beyond the categories proposed in the paper to accommodate metaphor-related tasks in diverse cultural contexts.
>
> Concerning Question B, this is not a broad range for the task of this paper. The 13 classifications used in this paper closely align with the source and target domain categories mentioned in the Metaphor Master List (MML)(Lakoff, 1994) at a fine-grained level. For example, in the MML, the sentence "She is free like a butterfly" is categorized as the target domain of behavior and the source domain of animals. The use of these 13 categories effectively guides the model to capture the conflicts between the source and target domains. However, excessively detailed classification would lead the model to rely on specific combinations of categories for inference, thus affecting its generalization capability.
>
> Concerning Question C, the annotators, including master's students and the authors themselves, have research backgrounds in metaphor computation and metaphor linguistics. Due to the anonymity of the authors and related personnel, specific information regarding their backgrounds has not been disclosed.
>
> Concerning Question D, we provided explanations and training to the annotators before annotation, along with detailed annotation guidelines. However, due to space limitations, this was not mentioned in the document.The following are some rules for determining metaphors in the annotation guidelines. We will include the complete content in the next version.
> Metaphor: We classify samples as metaphors when they exhibit the following characteristics. Objects within the image undergo modifications that highlight a particular feature, such as changes in shape, abnormal sizes, or unusual colors. Unconventional combinations of objects are observed in the image. The text utilizes techniques like homophones or allusions to emphasize a specific feature. There is a lack of harmony between the objects in the image and the objects mentioned in the text. Additionally, textual deformations are present within the image(Zhang et al., 2021).
> Non-metaphor: Samples classified as non-metaphor only depict the representation of objects in the image-text pairs without any implicit or deeper meanings.
>
> Concerning Question E, the values of n, p, and q represent the respective quantities of vocabulary terms in Set_macro , Set_micro ,Set_entity. The relevant explanations can be found in lines 505-511 of the original text.
>
> Concerning Question F, our initial collection of images included a significant number of images without accompanying text. However, we applied filtering techniques to remove these images systematically, employing both automated and manual methods to exclude those that lacked any textual content. Please refer to line 293 of the original text.
>
> Concerning Question G, this sentence explains why our model performs poorly on target domain and source domain classification tasks. The model struggles to accurately capture all features of the 13 categories, resulting in its performance being inferior in target domain and source domain classification tasks compared to metaphor identification tasks.
>
> Concerning Question H, yes, the model can learn richer features from a larger number of samples, which enables it to achieve better performance. We conducted a reanalysis of experimental data and found that the model demonstrates superior performance in categories with a larger sample size, such as artifact and natural object categories, compared to categories with a smaller sample size, such as relation and location categories.
>
> Concerning Question I, in multimodal image-text pairs, metaphors are typically expressed through the combination of objects from different domains. For example, in the picture, cigarettes are depicted as the lungs of a person, implicitly conveying the harm of smoking to health.
>
> Concerning the first reason for rejection, in the appendix, we present examples of both metaphorical and non-metaphorical images. For instance, as shown in Figure 7, the two images in the upper row are non-metaphorical. In the lower row, the image on the left is metaphorical. The text "毒" (poison) and "刑" (punishment) in the image conveys the metaphorical concept of drug injection and its consequences. In this image-text pair, the target domain is represented by a needle, while the source domain is symbolized by a rope. Similarly, the image on the right is metaphorical, with the text referring to a popular face mask. In this image-text pair, the target domain being a face mask and the source domain being water. It should be noted that not all the images in the appendix are metaphoric. In the upcoming version, we will include English translations and explanations for better comprehension.
>
> [1] George A Miller. 1995. Wordnet: a lexical database for english. Communications of the ACM, 38(11):39–41
> [2] George Lakoff. 1994. Master metaphor list. University of California.
> [3] Dongyu Zhang, Minghao Zhang, Heting Zhang, Liang Yang, Hongfei Lin: MultiMET: A Multimodal Dataset for Metaphor Understanding. ACL/IJCNLP (1) 2021: 3214-3225

---

### Meta-Review · Area_Chair_cZ4h · 2023-09-13

**Recommendation:** 4

**Metareview:**

The focus of the paper is the creation of a multimodal dataset for metaphor understanding. It’s main contributions are:

- a Chinese multimodal (images-texts) metaphor database from advertisements, annotated with metaphors, source and target domains of metaphors, and conveyed sentiments.

- a 4 domain lexicon;

- benchmark results for metaphor detection, source and target identification, and conveyed sentiment identification.

Reviewers generally appreciated the work but found some important weaknesses in the paper. The authors' responses were accurate and detailed, providing clarifications and the required details for many issues raised. While the required revisions are not substantial, they are indeed necessary (please refer to the Cons. section for specifics); anyways, the work is worthwhile.

**Pros**

- the paper describes the creation of a high-quality Chinese multimodal metaphor dataset, which constitutes a valuable resource for future researchers in the field of multimodal metaphor understanding;

- The extraction of data from creative advertisements and artistic works is an interestingly efficient method for creating useful data for metaphor understanding and yields valuable insights;

-  the domain classification approach followed enhances comprehension of metaphorical expressions across diverse data forms, and seems to align with human cognitive patterns for problem-solving;

- The paper is well-written, the argumentation easy to follow and understand;

- It presents interesting benchmark results;

- the work entails substantial effort and resources.

**Cons**

-  the paper leaves doubts about the choice of the level of granularity for identifying source and target domains. Some discussion seems to be needed in the paper, to help the reader understand the choices made;

- the description of the annotation process lacks details esp. on the annotators’ background, apparently for anonymity reasons. It is recommended to include this in the next version of the paper;

- a detailed analysis or discussion of the dataset and the annotation results is missing: e.g. analysis and discussion of disagreements, discussion on the observed properties of the multimodal metaphors; genre specific observations;

- Most appendix material is in Chinese-only. Translation in English is required.

---

### Decision · Program_Chairs · 2023-10-07

**Decision:**

Accept-Findings

**Comment:**

The focus of the paper is the creation of a multimodal dataset for metaphor understanding. It’s main contributions are:

- a Chinese multimodal (images-texts) metaphor database from advertisements, annotated with metaphors, source and target domains of metaphors, and conveyed sentiments.

- a 4 domain lexicon;

- benchmark results for metaphor detection, source and target identification, and conveyed sentiment identification.

Reviewers generally appreciated the work but found some important weaknesses in the paper. The authors' responses were accurate and detailed, providing clarifications and the required details for many issues raised. While the required revisions are not substantial, they are indeed necessary (please refer to the Cons. section for specifics); anyways, the work is worthwhile.

**Pros**

- the paper describes the creation of a high-quality Chinese multimodal metaphor dataset, which constitutes a valuable resource for future researchers in the field of multimodal metaphor understanding;

- The extraction of data from creative advertisements and artistic works is an interestingly efficient method for creating useful data for metaphor understanding and yields valuable insights;

-  the domain classification approach followed enhances comprehension of metaphorical expressions across diverse data forms, and seems to align with human cognitive patterns for problem-solving;

- The paper is well-written, the argumentation easy to follow and understand;

- It presents interesting benchmark results;

- the work entails substantial effort and resources.

**Cons**

-  the paper leaves doubts about the choice of the level of granularity for identifying source and target domains. Some discussion seems to be needed in the paper, to help the reader understand the choices made;

- the description of the annotation process lacks details esp. on the annotators’ background, apparently for anonymity reasons. It is recommended to include this in the next version of the paper;

- a detailed analysis or discussion of the dataset and the annotation results is missing: e.g. analysis and discussion of disagreements, discussion on the observed properties of the multimodal metaphors; genre specific observations;

- Most appendix material is in Chinese-only. Translation in English is required.